# A Brief Online Implicit Bias Intervention for School Mental Health Clinicians

**DOI:** 10.3390/ijerph19020679

**Published:** 2022-01-07

**Authors:** Freda F. Liu, Jessica Coifman, Erin McRee, Jeff Stone, Amy Law, Larissa Gaias, Rosemary Reyes, Calvin K. Lai, Irene V. Blair, Chia-li Yu, Heather Cook, Aaron R. Lyon

**Affiliations:** 1Department of Psychiatry and Behavioral Sciences, University of Washington School of Medicine, 6200 NE 74th Street, Suite 100, Seattle, WA 98115, USA; jcoifman@uw.edu (J.C.); emcree14@gmail.com (E.M.); rosereys@uw.edu (R.R.); hcookmed@uw.edu (H.C.); lyona@uw.edu (A.R.L.); 2Department of Psychology, University of Arizona, 1503 E University Blvd. Building 68, Tucson, AZ 85721, USA; jeffs@arizona.edu; 3Learning Gateway, University of Washington School of Medicine, 850 Republican St., Bldg. C-4, Seattle, WA 98109, USA; amylaw@uw.edu; 4Department of Psychology, University of Massachusetts, Lowell, 850 Broadway Street, Lowell, MA 01854, USA; larissa_gaias@uml.edu; 5Department of Psychological and Brain Sciences, Washington University in St. Louis, CB 1125, One Brookings Drive, St. Louis, MO 63130, USA; cklai4@gmail.com; 6Department of Psychology and Neuroscience, University of Colorado Boulder, Muenzinger D244, 345 UCB, Boulder, CO 80309, USA; irene.blair@colorado.edu; 7Department of Psychology, Pennsylvania State University, 140 Moore Building, University Park, State College, PA 16802, USA; cqy5287@psu.edu

**Keywords:** human-centered design, Implicit Association Test (IAT), implicit bias, mental healthcare professionals, online training, school mental health

## Abstract

Clinician bias has been identified as a potential contributor to persistent healthcare disparities across many medical specialties and service settings. Few studies have examined strategies to reduce clinician bias, especially in mental healthcare, despite decades of research evidencing service and outcome disparities in adult and pediatric populations. This manuscript describes an intervention development study and a pilot feasibility trial of the Virtual Implicit Bias Reduction and Neutralization Training (VIBRANT) for mental health clinicians in schools—where most youth in the U.S. access mental healthcare. Clinicians (*N* = 12) in the feasibility study—a non-randomized open trial—rated VIBRANT as highly usable, appropriate, acceptable, and feasible for their school-based practice. Preliminarily, clinicians appeared to demonstrate improvements in implicit bias knowledge, use of bias-management strategies, and implicit biases (as measured by the Implicit Association Test [IAT]) post-training. Moreover, putative mediators (e.g., clinicians’ VIBRANT strategies use, IAT *D* scores) and outcome variables (e.g., clinician-rated quality of rapport) generally demonstrated correlations in the expected directions. These pilot results suggest that brief and highly scalable online interventions such as VIBRANT are feasible and promising for addressing implicit bias among healthcare providers (e.g., mental health clinicians) and can have potential downstream impacts on minoritized youth’s care experience.

## 1. Introduction

Clinician bias has been identified as a potential contributor to persistent racial/ethnic disparities in healthcare across a wide range of medical specialties and service settings [1,2,3,4,5,6,7,8,9]. Systematic reviews of studies on bias among healthcare workers have found that physicians, nurses, and other healthcare professionals hold implicit racial biases much like the general population. Furthermore, these biases have been associated with differences in clinical care and patient experience [10,11,12,13,14]. Implicit bias includes *prejudice* (positive/negative evaluations associated with a given social group) and *stereotyping* (associating specific traits with a given social group) that operate automatically and are often only loosely related to explicit egalitarian beliefs [10,15,16]. Evidence for both types of implicit racial bias is abundant in the literature. Healthcare providers, regardless of specialty area (e.g., internal medicine, primary care, cardiology, emergency medicine, pediatrics, surgery, and psychiatry), profession (e.g., physicians, nurses, genetic counselors, occupational clinicians, mental health professionals), and training level (e.g., students, residents, fellows, faculty, and licensed professionals), have been found to have more negative and less positive implicit attitudes toward people of color (with the vast majority of the literature focusing on Black people) compared to non-Hispanic White people (i.e., pro-White bias) [4,7,9,10,11,12,17]. Research shows that these pro-White/anti-Black biases even extend to children and pediatric care [3,14,18,19,20]. Moreover, these biases have been found to be associated with lower quality of care [6,7,10,21].

### 1.1. Implicit Bias Interventions

While researchers have been studying implicit racial bias in healthcare for over two decades, development of interventions to address biases among healthcare professionals is still in its infancy [2]. In the broader implicit bias intervention literature, a recent meta-analysis identified nearly 500 studies that tested efforts to change implicit bias and found that implicit bias typically yielded with small to moderate effect sizes [18]. Consistent with findings from other recent studies, the meta-analysis found that the most effective strategies include those that (1) promote anti-bias goal setting (e.g., priming egalitarian norms and setting intentions to reduce biased responding) and proactive responding [22], (2) increase empathy (e.g., perspective taking and seeking commonality) [23,24], and (3) directly address the cognitive process of stereotyping (e.g., counter-stereotyping) [25,26,27,28]. However, the predominant majority of these interventions were decontextualized, one-time laboratory procedures with time-limited effects [29]. The few intervention programs that demonstrated meaningful impact on subsequent behavior and longer-term effects have several important features in common [24,30,31]. They (1) are highly contextualized such that the intervention is designed for a specific audience (e.g., university STEM faculty and medical students) to reduce bias in an identified process (e.g., patient interactions and hiring/promotion); (2) increase participants’ knowledge and awareness of their own implicit bias; (3) increase empathy skills (e.g., perspective gaining and seeking commonality); and (4) teach specific strategies that target the cognitive processes involved in stereotyping (e.g., counter-stereotyping and individuation). For these interventions, bias reduction effects have been documented up to 8 to 12 weeks after initial training [32] with broader systemic-level change up to 2 years post-intervention [33]. 

In terms of implicit bias intervention research with healthcare providers, one study found that a perspective-taking exercise effectively neutralized Black–White disparities in advanced nursing students’ pain-management recommendations [34] and another study found that a perspective-taking instruction given to third year medical students prior to clinical skills examination encounters improved Black standardized patients’ satisfaction ratings [35]. More recently, Chapman and colleagues [24] pilot tested a multi-session (approx. 4+ hours) intervention designed to increase empathy and understanding of the Latinx patient experience and found intervention effects on increasing empathy and reducing implicit bias toward Latinx patients among white medical students [24]. In a larger scale study, Stone and colleagues tested a multi-session (150 min total) in-person workshop with first year medical students and found reductions in implicit stereotyping of Latinx patients as “non-compliant” 3–7 days post-intervention [31]. One study with mental healthcare trainees examined the effects of a 15 week multi-cultural competence course and found it to have modest effects on reducing counseling graduate students’ implicit bias [36]. Results of these studies suggest that healthcare providers’ implicit bias is likely amenable to intervention and can improve after brief training. Noted limitations of these studies with healthcare providers include lack of follow-up (e.g., no data on whether intervention effects endure over time) and data on practicing professionals (participants were all students or trainees), highlighting the need for intervention studies in real-world settings with practicing professionals and longer-term follow-up. 

One important group of healthcare workers that has been previously understudied is mental health professionals (MHP). The few studies that examined racial bias among MHP (mostly trainees) found that they hold pro-White/anti-Black or anti-Latinx attitudes at similar levels to other healthcare providers and the general population [37,38]. Other evidence suggests that MHPs tend to be more empathetic compared to others [39], suggesting that their bias could be more amenable to intervention. However, bias among individuals who are expected to be more empathetic can also be more damaging to minoritized care recipients. Moreover, MHP’s bias may also have greater influence on downstream disparate outcomes given the mental health field’s heavy reliance on provider discretion [40] and limited adoption of treatment algorithms or standardized guidelines of care that could mitigate the impact of clinician bias [21]. Thus, developing implicit bias interventions for MHP may be particularly fruitful in terms of public health impact on existing disparities. 

### 1.2. Current Studies

The current studies (1) describe the iterative development and (2) pilot feasibility trial of a brief, interactive, online implicit bias intervention known as Virtual Implicit Bias Reduction and Neutralization Training (VIBRANT) with school-based mental health clinicians. School is the most common setting in which youth receive mental health services. It is also a low-resource, non-specialty service setting, where clinicians are often working with few local peers (e.g., being the only mental health provider in the building), and limited access to training and professional development. Moreover, school-based mental healthcare has been shown to ameliorate some service access disparities, while treatment engagement disparities persist [41], suggesting an impact of clinician bias. Thus, efficient and accessible online implicit bias trainings for clinicians may be particularly relevant for the school mental health context. The online training format not only reduces barriers and costs of training (e.g., travel time and arranging coverage for time away), especially for clinicians who are not all co-located (e.g., in different school buildings), but can also reduce common resistance to implicit bias training such as shame or embarrassment when considering one’s own bias alongside colleagues. These advantages of self-paced online implicit bias training make them more accessible and scalable than in-person workshops. 

Aims for the current studies: (1) The intervention development study aimed to produce a highly usable, brief, online training following an established human-centered design approach. (2) The pilot feasibility trial aimed to (a) assess the usability, acceptability, appropriateness, and feasibility of VIBRANT for implementation with school mental health clinicians as well as (b) the research feasibility of our measures, and recruitment and data collection protocol in preparation for an adequately powered randomized controlled trial. Both studies were conducted with approval by the Institutional Review Board of the University of Washington in Seattle, Washington, DC, USA. 

## 2. Intervention Development Study

VIBRANT was iteratively developed following the Discover, Design, Build, and Test framework for developing behavioral health interventions leveraging human-centered design and implementation science principles and methodology (see Figure 1) [42]. This approach has been refined through its application to 15 separate studies since its publication across a variety of health service interventions and delivery settings (e.g., primary care, specialty mental health, schools). We began by adapting core components of an empirically tested, live in-person implicit bias workshop originally designed for medical students and residents in primary care [31] into a self-paced interactive online training. Digital wireframes (storyboards) were initially developed by the first author to delineate the didactic content and learning activities of VIBRANT. This was then carefully reviewed with the developer of the original live training (co-author JS) to ensure accurate representation of critical components. Next, a fully functional digital prototype was developed by an instructional designer (co-author AL) and user tested with 6 school mental health clinicians (SMHC). Revisions to the prototype were made following feedback from the first round of user testing, and a second round of user testing with 6 more SMHC was conducted with this revised version of VIBRANT. Another round of revisions was completed to integrate feedback from Round 2 of user testing, and this final version was tested in a subsequent feasibility pilot study. The usability testing protocol was designed to assess users’ impression of the training’s navigability, key information or learning points via various interactions, overall usability, and any specific usability issues.

### 2.1. Materials and Methods

#### 2.1.1. Participants

In total, 12 school-based mental health clinicians (SMHC) who identified as non-Hispanic and White (NHW) were recruited from *local* school districts to participate in 2 separate rounds of user testing (6 SMHC per round). Human-centered design literature consistently recommends multiple, iterative tests with small samples since usability tests tend to yield diminishing returns after approximately 5 users [43,44,45,46]. Because VIBRANT was modeled after another brief online training for school mental health clinicians [47] that was developed and user-tested just months before, only 2 rounds of user testing were required, as core training activity types and navigation issues had already demonstrated good usability. 

Participant inclusion criteria required only that participants identified as NHW and routinely provide individual-level interventions or therapy to students in the K-12 setting and spend ≥50% of their time providing mental health services in schools. We intentionally recruited NHW participants to optimize interpretability of feedback recognizing that the target audience of the training is the population of predominately NHW SMHCs, generally consistent with the racial diversity within the broader behavioral health workforce in United States [48]. 

All 12 participants identified as female (consistent with local gender demographics of SMHC—85% female [49]) and held a master’s degree. Participants were between the ages of 18–64 with 83% identifying as between 25 and 44. Their experience as a SMHC ranged from being in their first to 20th year practicing in schools, with most participants (75%) in the 1st to 8th years range. Half of the participants indicated that Black and/or Latinx youth make up a substantial portion of their caseload (>20%). There were no notable differences between the two groups of SMHC who participated in Round 1 vs. Round 2 of usability testing.

#### 2.1.2. Procedures

Testing took place at a university research center, with two research team members (co-authors JC and EM)—one as lead facilitator/interviewer and the other as record keeper, who tracked user task completion, failures, bugs, navigation patterns, and time it took to complete each task. Participants were instructed to “think aloud” as they navigated through the digital prototypes and completed each activity. After each content section and learning activity, participants were asked to rate how easy/difficult the task was for [them] to complete” on a 1–5 Likert scale and then elaborate to explain their answer. On average, usability sessions were completed in approximately 90 min including a brief qualitative interview at the end. Sessions were audio recorded and reviewed to validate data collection accuracy.

#### 2.1.3. Measures

Participants also completed several quantitative measures including the Intervention Usability Scale (IUS) [50] at the end of the user testing session and pre-/post-session administrations of the Implicit Bias Knowledge Quiz (see below), and the Concern for Discrimination Scale [32].

##### The Intervention Usability Scale (IUS) 

The IUS [50] is a validated measure for assessing the usability of complex psychosocial interventions. It is adapted from the well-established System Usability Scale (SUS), which has been used in over 500 published studies [51,52]. The SUS was originally developed to assess usability of digital technology such as websites and software. It is a 10-item Likert scale rating measure that generates a score ranging from 0 to 100, with <50 being “unacceptable”, >70 “acceptable”, and >85 “excellent” usability. The IUS replaced the word “system” with the word “intervention” across all SUS items with no other adjustments. Sample items include, “I thought the intervention was easy to use.” And “I found the intervention unnecessarily complex.”

##### Implicit Bias Knowledge Quiz

This is a 10-item multiple choice test we developed for the current study to assess participants’ retention of key components of VIBRANT including definition of implicit bias, negative impact of clinician implicit bias, awareness of one’s own implicit bias, and the strategies to mitigate the influence of implicit bias and their application. See Appendix A for Knowledge Quiz Items and answers. 

##### Concern for Discrimination Scale (CfD)

The CfD is a previously published [32] 4-item Likert scale (1–10) measure that asks respondents to indicate their degree of agreement with statements such as “I consider racial discrimination to be a serious social problem” and “I’m not personally concerned about discrimination against Blacks” (reversed). We added parallel items referring to “Latinos” to mirror those that refer to “Blacks” so that 2 separate CfD scales could be calculated—one for CfD toward Black or African Americans, and one for Latinx Americans. The original study reported adequate internal reliability (*α* = 0.86). 

#### 2.1.4. Data Analysis 

Rapid qualitative data analysis was completed after each round of user testing to identify usability issues including any navigation problems and less than optimal user comprehension of key content. Quantitative data were reviewed to validate qualitative findings and assess overall usability and functionality of the training. Descriptive and comparison analyses were completed for the IUS, as well as pre-/post-knowledge quiz and CfD scores. We expected each successive version (alpha vs. beta) to be better in terms of number of usability issues identified and overall rating of usability. We expected participants’ implicit bias knowledge and concern for discrimination toward Black and Latinx people to increase from before to after the train.

### 2.2. Results

#### 2.2.1. Qualitative Findings

Usability issues identified in the first round of user testing focused on clarity of instruction for interactive learning tasks and feedback for correct/incorrect responses and progress, relevance and acceptability of examples, and overall visual appeal. Notably fewer issues were identified in Round 2 and the issues focused on accurate application of implicit bias strategies taught and the acceptability of case examples. Overall, comprehension of key information improved, and fewer navigation errors were observed from Round 1 to Round 2. Participants agreed that having learners complete the Implicit Association Test (IAT) to reflect on their own bias was an effective engagement strategy and made the training feel more personally relevant (e.g., “It was good to have the IAT, especially first, because it show[s] why this training is important from the beginning and it makes you more interested.” “I don’t consider myself racist, but then you take this test and [it shows that] you have a preference and it’s like, ‘damn’”). Participants also found the training to be engaging, relevant, and useful overall (e.g., “I was motivated and interested. It was engaging and user-friendly”. “It was much more interactive than other online trainings. You can see how these concepts are relevant to you as a clinician”. “It hits home and resonates. It’s important and a concrete way to [address bias], it fits with my values”).

#### 2.2.2. Quantitative Findings

As expected, usability improved from “acceptable” (average IUS = 78.3, *SD* = 22.17) in Round 1 to better than “excellent” (average IUS = 94.48, *SD* = 4.31) in Round 2, *Hedges’ g =* 0.94. While participants’ implicit bias knowledge improved from pre to post-training (*M* = 56.4%, *SD* = 18.04; *M* = 71.8%, *SD* = 11.68, respectively) for clinicians across both rounds of user testing with an average increase of 15.5% (*Hedges’ g =* 0.94), post-training knowledge quiz scores were 11% higher in Round 2 (average 78.0%, *SD* = 13.04) than in Round 1 (average 66.7%, *SD* = 8.16), *Hedges’ g =* 0.98. Across both rounds of usability testing, participants demonstrated increases in their concern for discrimination toward Black, *Hedges’ g =* 0.51, and Latinx, *Hedges’ g =* 0.62, youth.

#### 2.2.3. The VIBRANT Intervention

Upon integration of user feedback from the second round of user testing, a “gold” version of VIBRANT was tested in a small pilot feasibility trial (see below). This version of the VIBRANT intervention is outlined in Box 1 and described here. Sections 1–3 consists of introduction and didactic information including learner engagement activities (e.g., content that makes the training specifically relevant to SMHC). Section 4 contains the primary behavior change components of the training. In this section, specific bias-management strategies that address both implicit prejudice and stereotyping (see Box 1) are presented through case-based learning activities. Clinicians are introduced to hypothetical racial/ethnic minority adolescent patients and given opportunities to practice each of the bias-management strategies by “interacting” with the patient (including first-person perspective visuals to simulate Black and Latinx youths’ points of view, which has been shown to decrease implicit bias [53]. The specific bias management skills include (a) seeking commonality, a way for clinicians to identify what they have in common with a patient who does not share their racial/ethnic identity to reduce intergroup distance (and experiencing those different from us as “other”); (b) perspective gaining, a strategy to help clinicians understand the situation from the patient’s vantage point taking into account the totality of patient’s history and context to increase empathy; and (c) counter-stereotyping with individuation, which involves recognizing salient stereotypes relevant to a minoritized patient (e.g., “Black youth are prone to delinquency”) and proactively seeking stereotype-challenging information (e.g., which problem behaviors has this patient actually engaged in?). This process neutralizes the heuristic function of stereotypes by helping clinicians (1) challenge inaccurate stereotypic assumptions and (2) learn more about the patient they are working with, which helps individualize them from their identified group (e.g., Black youth) [54]. Finally, the training wraps up with an activity to strengthen clinicians’ implementation intention by asking them to make a semi-public plan (e.g., emailing a colleague) for using these skills to address implicit bias within their clinical practice.

Box 1Outline of VIBRANT Components.
**VIBRANT Training Components**
**1. Learner engagement strategies & introduction** (5 min)
Review SMH clinicians’ existing relevant skillsHighlight relevance of training to clinical practiceIntroduce prejudice & stereotyping w/a personally relevant example—stereotypes of “SMH counselors”
**2. Didactic material to normalize implicit bias**
Evolutionary function, ubiquity (5 min)Automaticity of implicit prejudice & stereotyping (5 min)
Photo observation taskStroop Task (color naming)
**3. Implicit Association Test** (7 minute)
Learner completes Race IAT & receives feedbackAddressing the bias blind spot (Pronin et al., 2002) [55]
**4. Bias management strategies (case-based learning)**
Seeking-commonality (5 min)Perspective gaining (5 min)Counter stereotyping & individuation (10 min)
**5. Strengthen Implementation Intentions** (3 min)
Make public commitment (email colleagues)

## 3. Feasibility Pilot Study

A non-randomized open feasibility trial using a standard pre–post design was conducted with a small but representative sample of SMHCs from across the US to assess VIBRANT’s usability, acceptability, appropriateness, and feasibility for implementation with school mental health clinicians. This pilot study also aimed to assess the suitability and feasibility of our research procedures (including measures) for a subsequent larger trial. Thus, in addition to collecting participant ratings on VIBRANT’s usability, acceptability, appropriateness, and feasibility, we also assessed clinicians’ implicit bias knowledge pre- and post-training, and implicit prejudice and stereotyping toward Black and Latinx Youth using the Implicit Association Test (IAT) throughout the study period (up to 24 weeks post-training). We collected clinicians’ self-reported use of VIBRANT debiasing strategies with patients on their caseload and clinicians’ perceptions of rapport with each of their patients at 2, 6, 14, and 24 weeks after they completed the training. While this study was not designed or powered to evaluate VIBRANT’s overall effectiveness on reducing implicit bias or its downstream impact, we included these measures on putative mediators per our theory of change (see Figure 2), primarily to assess the appropriateness of these measures for a larger trial. As depicted in Figure 2, we hypothesized that participants would evidence improved implicit bias knowledge and sustained use of VIBRANT bias-management strategies after training. We expected implicit bias knowledge (as evidenced by their performance on an objective quiz) and frequency of VIBRANT bias-management strategies use to be negatively associated with measures of implicit bias (i.e., IAT *D* scores). In turn, we expected measures of implicit bias to be negatively associated with clinicians’ ratings of rapport with patients on their caseload. We also expected rapport to improve with time and explored whether that would differ for Black and Latinx youth (BLY) vs. non-BLY. Conceptually, this means that better implicit bias knowledge and more frequent VIBRANT strategies use should be associated with indicators of lower implicit bias, which should be associated with better rapport, especially with Black and Latinx youths on clinicians’ caseloads.

### 3.1. Materials and Methods

#### 3.1.1. Participants

In keeping with recommendations for pilot feasibility trials [56,57], we recruited a small but representative sample (*N* = 12) of SMHC from the control group of a larger study examining the impact of brief online training for measurement-based care [47]. The current sample size is comparable to that of previous studies with similar aims (e.g., [58]) and sufficient to assess the implementation outcome variables (i.e., usability, acceptability, appropriateness, feasibility) and research feasibility with adequate precision to inform the decision to progress to a larger trial [57]. 

Participants of the parent study were originally recruited from across the United States via professional listservs and social media. Inclusion criteria required that participants routinely provide individual-level interventions or therapy to students in the K-12 setting and spend ≥50% of their time providing mental health services in schools. There were no racial/ethnic identity requirements for this study, and the only exclusion criteria was participation in the intervention development study. Demographic breakdown of the current study sample is similar those reported in previous studies with school mental health clinicians in the United States [59,60]. All participants identified as female; 10 of the 12 clinicians identified as “White”, 1 as White and other (Middle Eastern), and 1 as Asian. Half of the participants (*n* = 6) were from the Midwest, 3 were from the South, 2 from the Northeast, and 1 from the West Coast. They ranged in age (75% identified as between 25–44), years of experience in school mental health (75% indicated they were within their 2nd to 8th year). One participant dropped out after completing the post-training assessment. Thus, while pre-/post-training data were obtained from all 12 participants, follow-up data were only available for 11 participants. 

#### 3.1.2. Procedures

Because VIBRANT is entirely online (as were all study assessments), all SMHC participated in this study remotely from across the United States. Participants were asked to complete the parent study online training (~60 min) first then VIBRANT (~45 min). After completing baseline measures, participants were given 2 weeks to complete both training and post-training assessments. Participants completed follow-up assessments at 2, 6, 14, and 24 weeks after training. Participants completed a battery of self-report measures at each assessment timepoint in addition to 4 distinct Implicit Association Tests (see Measures) that we developed for the current study and a caseload log that documented their use of VIBRANT strategies and their perception of rapport with each patient seen during the study period. See Table 1 for measure administration schedule.

#### 3.1.3. Measures

In addition to the demographics form, the Intervention Usability Scale, the Concern for Discrimination Scale, and the Implicit Bias Knowledge Quiz, described above, the following measures were also administered throughout the study period. 

##### Measures of Acceptability, Appropriateness, and Feasibility

A set of three 4-item Likert scales [61] were administered post-training to assess participants’ perception of VIBRANT—Acceptability of Intervention Measure (AIM), Intervention Appropriateness Measure (IAM), and Feasibility of Intervention Measure (FIM). Sample items include “I like this training” (AIM), “This training seems fitting” (IAM), and “this training seems doable” (FIM). The AIM, IAM, and FIM, have demonstrated a 3-factor structure with acceptable fit (CFI = 0.96, RMSEA = 0.08), internal reliability (α = 0.85–0.91), test–retest reliability (0.73–0.88), and construct validity. Study on predictive validity and benchmarking are underway, such that practical cutoffs of acceptable ratings are not yet available. Therefore, the current study used an overall rating of 80% or higher (average ratings >4 on a 5-point Liker scale) as an indicator of acceptable levels of these constructs.

##### Caseload Service Logs

Participants completed caseload service logs at 2, 6, 14, and 24 weeks post-training. They logged every patient seen during the prior 2 week period (with a pseudonym) along with patient demographic information (gender, age, grade, race, and ethnicity), and indicated which if any of the 3 main VIBRANT bias-management strategies (i.e., seeking commonality, gaining perspective, and counter-stereotyping) were used for each identified patient. Scores on each of the three bias-management strategies were added together to create an overall score representing the use of each of the strategies (0–3) for each client. Clinicians also rated their perception of rapport using a Likert scale from 0 to 10 (0 = cold, disconnected, uncomfortable; 5 = mediocre, weak connection; 10 = warm, connected, very comfortable) with each patient during the session captured on their service log. 

##### Implicit Association Test (IAT)

The IAT is the most well-validated measure of implicit attitudes [62,63]. It is a computerized categorization task that measures the relative strength of associations by examining the speed with which people sort images/words to categories. For example, people who have stronger pro-White/anti-Black biases would be faster at pairing White people with positive attributes and Black people with negative attributes than the reverse (i.e., Black + positive/White + negative). This difference in reaction times can be computed as an IAT *D* score, where a positive score indicates a pro-White/anti-minoritized group (Black in this example) bias, a negative score indicates a pro-minoritized group/anti-White bias, and a zero or near-zero score indicates no detectable bias [64]. 

At each assessment timepoint, participants completed 4 IATs that our team developed and validated in a previous study [65]. Two of the IATs assess implicit prejudice (e.g., Black–White/good–bad and Latinx–White/good–bad) and two assess implicit stereotyping (i.e., Black–White/defiant and obedient; Latinx–White/academic failure vs. success) of Black and Latinx youth. The target stereotypes used in these IATs were empirically identified in a previous study with SMHCs [65]. Each IAT takes approximately 6 min to complete; participants were encouraged to take breaks between each IAT to limit fatigue. 

#### 3.1.4. Data Analysis

##### Data Validation and Identifying Outliers

Raw data were checked for consistency and meaningful responding (e.g., not selecting the same rating for all items). Outliers were identified as scores more than 2.5 standard deviations away from the mean and excluded [66]. Given the relative impact of each score in small sample sizes, analyses were conducted with and without outliers to ensure that exclusions did not change results in meaningful ways. Generally, we took a maximum information approach and retained as much data as possible across all analyses. Because we did not have a comparison group, the impact of history or cohort effects cannot be ruled out. The 24 week assessment timepoint occurred after COVID-19 was announced as a global pandemic on March 11, 2020 and schools shut down across much of the United States. Although we report on our 24 week data for the sake of completeness, these results must be interpreted with caution.

##### Clinician Perception of Usability and Feasibility

Descriptive analysis was completed on the IUS, AIM, IAM, and FIM, to assess clinician-reported usability, appropriateness, acceptability, and feasibility of VIBRANT. Mean scores were compared to pre-established benchmarks to determine whether VIBRANT met expected usability (IUS > 70) and feasibility (AIM, IAM, FIM > 4) standards. 

##### Clinicians’ Implicit Bias Knowledge 

Given that the knowledge quiz was designed to cover the different topics and constructs in VIBRANT, traditional measures of internal consistency (e.g., Cronbach’s alpha) seems inappropriate for items that were not expected to represent a single cohesive construct. We did assess test–retest reliability by calculating the intraclass correlation (ICC) between the knowledge quiz score immediately and 14 weeks post-training. Following published guidelines for assessing reliability with ICCs [67], we calculated an ICC with a two-way mixed-effects model with an absolute agreement definition using IBM SPSS Statistics for Windows, version 28 [68].

Descriptive and comparative analyses were completed on scores of the Implicit Bias Knowledge Quiz at pre-, post-, 14 and 24 weeks post-training. Given the small sample size and exploratory nature, these analyses focused on estimating effect sizes and observing trends as opposed to null hypothesis significance testing. *Hedges’ g* (instead for *Cohen’s D*) was used given that it provides more accurate estimates for small (*N* < 20) samples [69]. Conventions for interpretation for *Hedges’ g* is similar to *Cohen’s D*.

##### Clinicians’ Use of De-Biasing Skills and Perceived Rapport with Students

Descriptive analysis of caseload reports was conducted to assess how often clinicians used each of the three primary debiasing strategies they learned in VIBRANT (i.e., seeking commonality, perspective gaining, and counter-stereotyping) with their patients. In total, clinicians reported on 306 patients, with an average of 27.8 patients per clinician. Of the 297 patients with race/ethnicity data reported (2.9% missing), 162 (52.9% identified as Black or Latinx and 135 (44.1%) identified as a racial/ethnic category other than Black or Latinx. On average, patients were 12.98 (SD = 2.95) years old; 169 (55.2%) identified as male and 137 (44.8%) identified as female. 

Unconditional longitudinal growth models were estimated for clinicians’ use of VIBRANT bias-management strategies composite and rapport with each of their patients as recorded on their caseload logs at 2, 6, 14, and 24 weeks post-training with the slope loadings fixed to 0, 4, 8, and 12 representing the number of weeks between each assessment; thus, a 1 unit change in time represents 1 week. After the initial longitudinal models were estimated, a multi-group analysis was applied to examine whether the growth in strategies use and rapport differed for BLY, as compared to all other students. These groups were chosen due to VIBRANT’s focus on reducing clinicians’ stereotypes regarding BLY specifically; however, small cell sizes prevented further disaggregated analyses. Wald tests were used to examine whether the intercept and slope parameters were significantly different across the two groups. All models were estimated using Mplus 8. Full information maximum likelihood was used to handle missing data. 

##### Clinicians’ Implicit Bias

Average Clinician IAT scores across all available assessment timepoints (pre-, post-, 2, 14 and 24 weeks post-training) for each of the 4 IATs were compared and observed for notable changes and trends. 

##### Exploring Mechanistic Relationships

Cross-sectional correlation analyses were conducted to explore theoretically anticipated relationships among study variables at the most logical assessment timepoint per our model of change (Figure 2). Specifically, we examined correlations between implicit bias knowledge quiz score and all 4 IAT *D* scores post-training. At 2, 14, and 24 weeks post-training (all available assessment points), we examined correlations between the average frequency of VIBRANT strategies use with Black Youth on each clinician’s caseload with clinician’s D scores of the Black/White–Good/Bad and Black/White–Obedient/Disobedient IATs; and average frequency of VIBRANT strategies use with Latinx youth on clinician’s case load with clinician’s *D* scores on the Latinx/White–Good/Bad and Latinx/White–Academic success/failure IATs. At the same timepoints, we examined correlations between the two Black/White or Latinx/White IAT *D* scores and clinician’s average rating of rapport with Black or Latinx youth on their caseload, respectively. Given the small sample size, we avoided interpreting individual correlation coefficients and focused on general patterns of associations. Ranges of effect sizes in terms of Pearson *r* (as opposed to significance testing) are reported when appropriate. 

### 3.2. Results

#### 3.2.1. Usability and Feasibility 

One participant’s scores were identified as outliers on all four usability and feasibility measures and excluded per the analysis plan (Usability and feasibility ratings all met benchmarks even when the low outlier score for each scale was included). Remaining IUS scores yielded a mean usability rating in the “good” to “excellent” range (IUS Mean = 80.23, *SD* = 10.63). Similarly, VIBRANT was rated to be highly appropriate (*M* = 4.43, *SD* = 0.51), acceptable (*M* = 4.43, *SD* = 0.51), and feasible (*M* = 4.47, *SD* = 0.48) for school mental health clinicians, with all average ratings above our pre-established benchmark of 4 (on a 5-point scale).

#### 3.2.2. Implicit Bias Knowledge 

ICC analysis indicated moderate test–retest reliability (ICC = 0.68), which was considered acceptable given the long (12 weeks) lag time between the 2 administrations of the quiz. The same participant with outlier scores on the usability and feasibility measures yielded an outlier score on the post-training administration and had no follow-up data on the implicit bias knowledge quiz and was thus excluded from longitudinal comparisons on this measure. Among the remaining 11 participants, mean implicit bias knowledge quiz score notably improved from pre- (64.5%) to post-training (80.9%), yielding a 16.4% increase (*Hedges’ g* = 1.49). Participants also appeared to have retained their implicit bias knowledge gains, as their follow-up quiz scores at 14 and 24 weeks post-training remained high at 78.2% and 80.0%, respectively. 

#### 3.2.3. Use of VIBRANT Bias-Management Strategies and Rapport

Summary statistics revealed that 2 weeks post-training, clinicians used seeking commonality (80.5%) and perspective gaining (76.4%) with the majority of patients on their caseload, but only used counter-stereotyping with a few patients (13.4%). While clinicians’ use of counter-stereotyping improved overtime (32.3% in week 14 at its peak), it was never used as frequently as the other two strategies (*Hedges’ g* = 1.28, 1.52). See Figure 3. 

#### 3.2.4. Growth in Use of De-Biasing Strategies and Rapport over Time

Both the strategies (χ^2^ (8) = 12.99, *p* = 0.11; CFI = 0.96; RMSEA = 0.05, 90% CI (0.00, 0.09); SRMR = 0.10) and rapport (χ^2^ (8) = 13.07, *p* = 0.11; CFI = 0.99; RMSEA = 0.05, 90% CI (0.00, 0.09); SRMR = 0.10) growth models demonstrated adequate fit to the data. Table 2 presents clinician’s reports of their strategy use and rapport at each data collection point and the intercept and slope of the growth models for each variable for the overall sample and both BLY and non-BLY. In the overall sample, clinicians reported a significant increase in the use of VIBRANT bias-management strategies and rapport over 24 weeks. Multi-group analyses suggested that the intercepts should be constrained across groups for both strategies and rapport (*p*s > 0.64). In addition, these analyses indicated that the slope parameter should remain constrained between groups for the strategies model (*p* = 0.97) but freed for the rapport model (*p* = 0.01). For both Black/Latinx youth (BLY) and non-BLY youth, there was an equivalent, significant increase in the number of VIBRANT bias-management strategies clinicians reported using over the course of treatment. There was a significant increase in clinicians’ reported rapport with BLY youth over the course of treatment, but no significant growth in clinicians’ rapport with the non-BLY youth.

#### 3.2.5. Clinicians’ Implicit Bias

At baseline, participants evidenced the strongest bias on the Black/White–Good/Bad IAT (Mean *D* score = 0.44, *SD* = 0.24), which decreased post-training (*M* = 0.31, *SD* = 0.41, *Hedges’ g* = −0.28), and continued in a generally downward trend through 2, 14, and 24 weeks post-training (greatest *Hedges’ g* = −0.38). While participants evidenced less bias on the other 3 IATs at baseline (*D* scores = 0.23 to 0.33, *SD* = 0.24 to 0.37, largest *Hedges’ g* = 0.67), these *D* scores showed slight increases (*D* scores = 0.26 to 0.38, *SD* = 0.28 to 0.43, largest *Hedges’ g* = 0.15) post-training. Nonetheless, they too evidenced a generally decreasing trend over time (see Figure 4).

#### 3.2.6. Exploring Mechanistic Relationships

Inconsistent with expectations, post-training implicit bias knowledge quiz score was strongly negatively correlated (*r* = −0.63) with the Black/White–Good/Bad IAT but not the other 3 IATs. All other correlations were weak (<|0.3|). As hypothesized, bias-management strategy use with Black and Latinx youth was generally negatively associated with the relevant IAT *D* scores, with most Pearson *r* coefficients in the moderate to large range >−0.35. Similarly, IAT *D* scores were generally negatively associated with clinician’s rating of rapport with the referent group (Black or Latinx youth), with half of the Pearson *r* coefficients in the moderate to large range >−0.35. See Table 3 for correlation coefficients.

## 4. Discussion

Using an iterative human-centered design approach to developing behavioral health interventions, we created a brief, interactive, online implicit bias training that school-based mental health clinicians found relevant, engaging, and useful. Clinicians seemed receptive to feedback about their own bias in this online training format and found it impactful on their understanding of implicit bias overall. The resulting training took participants, on average, 45 min to complete and covered psychoeducation on implicit bias as well as 3 of the most effective bias-management strategies documented in the extant literature: seeking commonality, perspective gaining, and counter-stereotyping with individuation [23,25,27,28]. This makes VIBRANT notably more efficient and scalable than previously published trainings for healthcare providers [24,31,38], which is particularly important in a post-pandemic world that is increasingly more reliant on remote (as opposed to in-person) training options. 

Data across the intervention development and feasibility study showed that participants found VIBRANT to be highly usable, appropriate, acceptable, and feasible to implement in the school context, and their implicit bias knowledge improved after completing VIBRANT and gains were maintained up to 6 months after training. The feasibility study found that after training, participants’ use of VIBRANT strategies increased and sustained over 24 weeks without any additional intervention, echoing findings of a previous study with a short follow-up period [32]. However, clinicians did not use the three bias-management strategies with equal frequency; they used the counter-stereotyping strategy with much less frequency than the other two strategies. Post hoc analysis revealed that although they did not consistently endorse counter-stereotyping as more difficult to use, they thought it would be less effective than the other two strategies for reducing implicit bias and improving their interactions with students of a background different from their own. Especially telling was that clinicians anticipated fewer opportunities to use counter-stereotyping than the other two strategies despite explicit instruction in the VIBRANT training on how all three strategies can be used any time they are interacting with people who they perceive to be from a different social group than those the clinicians identify with. This pattern of finding suggests that more work can be done to adjust the training content relevant to the counter-stereotyping strategy to improve its perceived effectiveness, increase clinician motivation to use, and improve overall uptake. These findings also highlight the importance of assessing participants’ perceptions, expectations, and use of bias-management strategies in intervention studies to better understand mechanisms of action for a given intervention. 

While this study was neither designed nor powered to assess the overall effectiveness of VIBRANT, we did collect and analyze data for research feasibility. Preliminary data from this feasibility open trial found a generally decreasing trend in average IAT *D* scores and improvements in clinicians’ perceptions of rapport with Black and Latinx youth on their caseload. It was somewhat surprising that rapport did not grow for non-BLYs. While it is tempting to interpret this differential growth pattern as a potential intervention effect, the Hawthorne effect (i.e., the impact of clinicians knowing that their caseload reports will be reviewed by researchers) [70] or other experimental effects (e.g., clinicians correctly guessing the hypothesis of the experiment and behaving/reporting accordingly) cannot be ruled out without a randomized control group. Nonetheless, this was an excellent feasibility test for this method of collecting case-level clinician behavior data—an efficient way to increase power when working with small samples.

Exploratory correlation analyses results provide some preliminary evidence on putative mechanisms of change. Specifically, higher frequency of clinicians’ self-reported bias-management strategies use with Black and Latinx youth on their caseload seems to be associated with lower IAT *D* scores (indicating less bias), and lower D scores appears to be associated with better clinician-rated rapport with Black and Latinx youth. This signal that lower clinician bias may be related to better patient outcomes is broadly consistent with results of previous studies and reviews [11,12,24]. In contrast, implicit bias knowledge was not consistently associated with IAT *D* scores. Together, these results suggest that at least bias-management strategies use and implicit bias as measured by the IAT may be important putative mechanisms of change in how VIBRANT ultimately impacts minoritized youth’s mental healthcare experiences and deserves further study. 

### 4.1. Limitations

Both studies reported here have small sample sizes, which are appropriate for the developmental stage of VIBRANT (i.e., intervention development and feasibility pilot), but not reliable for drawing firm conclusions from any inferential statistics or generalizing current findings to other contexts. The lack of a control group and randomized group assignment prevents any causal interpretations. Although the pre–post open trial design alludes to potential intervention effects, maturation and history effects cannot be ruled out. In fact, because the last assessment timepoint occurred after the announcement of COVID-19 as a global pandemic, which led to mass school closures and transition to remote learning across the United States, it is highly difficult to surmise the specific impacts on school mental healthcare and data collected during this time. Thus, results reported here involving week 24 data must be interpreted with caution. It is also worth noting that week 24 data collection occurred before the death of George Floyd and widespread anti-racism protests in the US and across the globe. Moreover, the reliance on clinician reporting (albeit with a variety of objective, records, and self-report measures) can also confound our results in that clinicians who completed an implicit bias training may be perceiving improvements in their rapport with minoritized youth simply because they are trying to curb their implicit bias and that youth involved in those interaction may not actually share the clinicians’ perspective.

### 4.2. Future Directions

Using Thabane and colleague’s approach to determining adequate feasibility for progression to a larger trial [71], current pilot findings clearly suggest continuing on to the next phase with minor modifications to the VIBRANT training—namely adjustments to activities for teaching the counter-stereotyping strategy to improve learner perceptions and usage. In the near future, we hope to conduct an adequately powered randomized controlled trial to test intervention effects of VIBRANT on reducing clinicians’ assessed implicit bias (via the IAT) and improving patients’ mental healthcare experience (via youth report), as mediated by proximal mechanism of implicit bias knowledge and use of bias-management strategies (as depicted in Figure 2). Future research may also evaluate the impact of VIBRANT on clinical decision-making (e.g., diagnosis, treatment recommendations) and provider-patient interactions as other critical mechanisms to patient outcomes. Because the core bias-management strategies covered in VIBRANT are broadly applicable, the training can also be easily adapted and scaled to benefit healthcare professionals from any number of specialty or generalist practice settings. 

## 5. Conclusions

To the best of our knowledge, VIBRANT is the first online implicit bias intervention developed for mental health clinicians. It is also more efficient and scalable than previous published trainings for healthcare providers [24,31,38]. Our feasibility study findings suggest that clinicians seem to be able to acquire and retain knowledge about implicit bias as well as adopt and sustain bias-management strategies in their clinical practice after a brief (~45 min), self-paced, interactive, skill-based, online training such as VIBRANT. While these studies were not designed to test VIBRANT’s effectiveness to reduce clinicians’ implicit bias or downstream impacts on minoritized youth patients’ experience in school mental healthcare, our findings clearly indicate adequate feasibility and sufficient signal to pursue a fully powered randomized controlled trial. The feasibility trial also highlights the importance of conducting implicit bias intervention studies in real-world settings with longer-term follow up. More broadly, these studies demonstrate the feasibility and public health potential of similarly efficient and scalable implicit bias interventions for healthcare providers.

## Figures and Tables

**Figure 1 ijerph-19-00679-f001:**
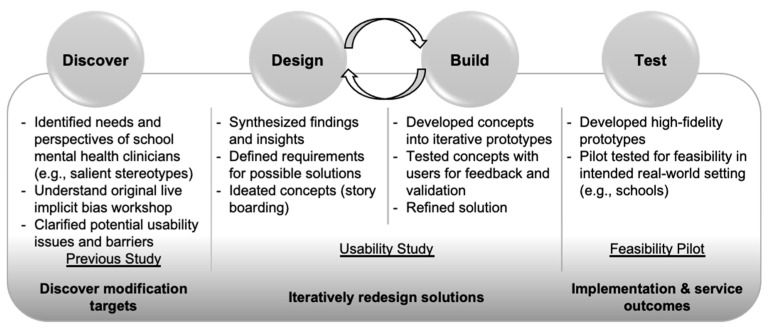
Discover, Design, Build, and Test Development Process Adapted from Lyon et al., 2019 [42].

**Figure 2 ijerph-19-00679-f002:**
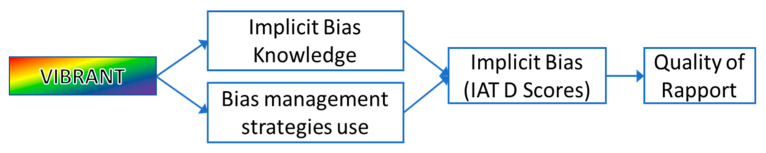
Theory of Change Model. IAT—Implicit Association Test.

**Figure 3 ijerph-19-00679-f003:**
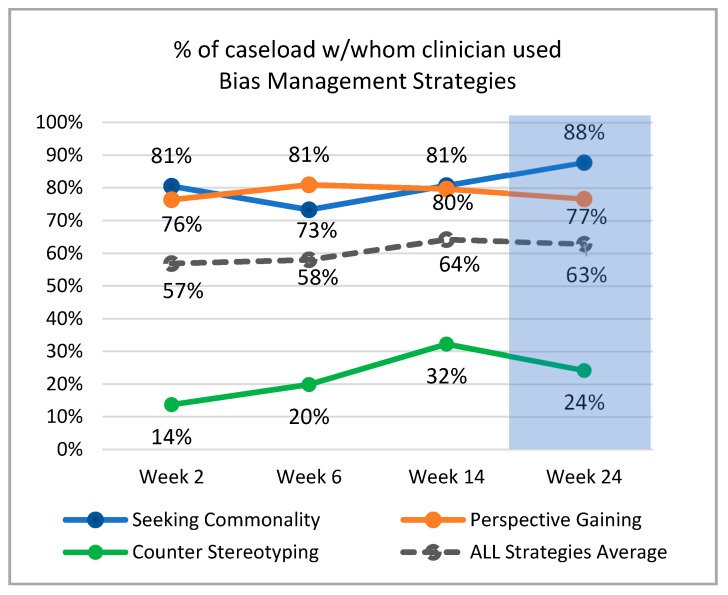
Average Frequency of VIBRANT Strategies Use across Clinicians’ Caseloads.

**Figure 4 ijerph-19-00679-f004:**
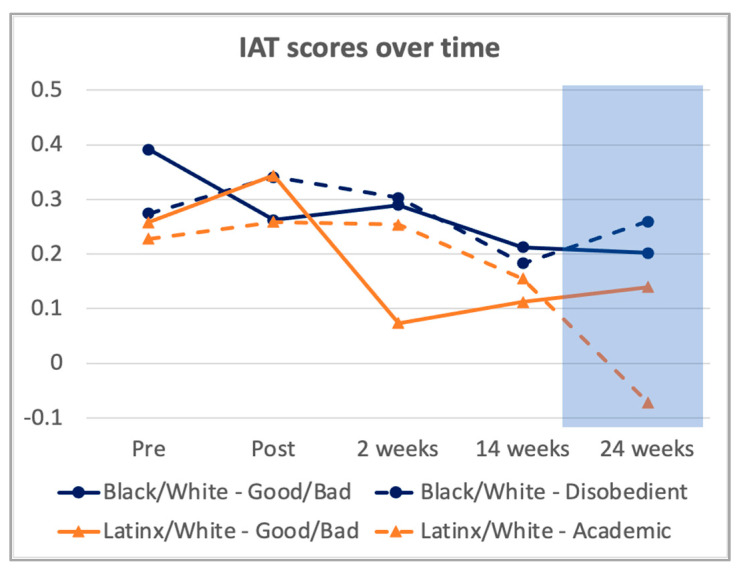
IAT Scores at Each Assessment Timepoint.

**Table 1 ijerph-19-00679-t001:** Measure Administration Schedule.

VIBRANT Data Collection	T1	T2	T3	T4	T5	T6
	Base-Line	Post	Week 2	Week 6	Week 14	Week 24
Clinician Demographics Form	X					
Intervention Usability Scale		x				
Acceptability, Appropriateness, Feasibility Measures		x				
Qualitative Feedback		x				x
Implicit Bias Knowledge Quiz	X	x			x	x
Implicit Association Test	X	x	x		x	x
Clinician Caseload Service Log			x	x	x	x

**Table 2 ijerph-19-00679-t002:** Clinician-Reported VIBRANT Strategies and Rapport by Race/Ethnicity.

	Overall Sample	Black/Latinx	Non-Black/Latinx
** *Strategies* **			
**Weeks Post-Training**	**M(SD)**	**M(SD)**	**M(SD)**
2 weeks	1.57 (0.79)	1.55 (0.91)	1.67 (0.62)
6 weeks	1.69 (0.92)	1.71 (1.01)	1.75 (0.81)
14 weeks	2.00 (0.84)	2.05 (0.73)	2.06 (0.92)
24 weeks	1.74 (0.83)	1.80 (0.80)	1.72 (0.85)
	**B(SE)**	**B(SE)**	**B(SE)**
Intercept	1.66(0.19)	1.64(0.18)	1.64(0.18)
Slope	0.04 (0.01) *	0.04(0.01) *	0.04(0.01) *
** *Rapport* **			
**Weeks Post-Training**	**M(SD)**	**M(SD)**	**M(SD)**
2 weeks	7.36 (1.68)	7.41 (1.71)	7.38 (1.51)
6 weeks	7.44 (1.67)	7.34 (1.77)	7.63 (1.58)
14 weeks	7.69 (1.57)	7.97 (1.53)	7.29 (1.58)
24 weeks	7.76 (1.46)	8.00 (1.63)	7.55 (1.52)
	**B(SE)**	**B(SE)**	**B(SE)**
Intercept	7.55 (0.37)	7.59 (0.38)	7.59 (0.38)
Slope	0.02 (0.01) *	0.03 (0.01) ***	0.01 (0.01)

Note: SE = Standard Error, * *p* < 0.05, *** *p* < 0.001.

**Table 3 ijerph-19-00679-t003:** Correlation Coefficients for Cross-Sectional Relationships between Strategies Use and IAT *D* Scores and Rapport and IAT *D* Scores.

** *Strategy Use* **			
**Implicit Association Test**	**Week 2**	**Week 14**	**Week 24**
Black/White_Obedient/Disobedient	−0.52732	−0.35456	−0.14593
Black/White_Good/Bad	−0.03476	0.252209	−0.40286
Latinx/White_Academic Failure/Success	−0.48102	−0.45801	−0.38922
Latinx/White_Good/Bad	−0.35073	0.046415	0.158649
** *Rapport w/Reference Group* **			
**Implicit Association Test**	**Week 2**	**Week 14**	**Week 24**
Black/White_Obedien/Disobedient	−0.68675	−0.76049	−0.27601
Black/White_Good/Bad	−0.03455	−0.36688	−0.12346
Latinx/White_Academic Failure/Success	−0.17831	−0.52114	−0.66884
Latinx/White_Good/Bad	−0.19024	−0.20642	−0.46028

## Data Availability

The data presented in this study are available on request from the corresponding author.

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
