# Peer review of "A Brief Online Implicit Bias Intervention for School Mental Health Clinicians"

_ijerph, 2022, doi:10.3390/ijerph19020679_

Round 1

Reviewer 1 Report

Although this manuscript deals with an interesting topic, it requires more theoretical support to defend the results. In this sense, it is necessary to increase the number of bibliographic references related to the subject and that they be from the last five years. In addition, the number of people who have participated is very low, which makes it difficult to generalize to other contexts. Greater emphasis should be placed on justifying why this number was used and why it is possible to conduct a study with this number. Finally, the discussion should "discuss" the results with those of other authors mostly seen in the introduction and the initial theoretical framework. As strong points of the manuscript, the information has been greatly refined and several variables of interest have been analyzed. Thank you for your attention. 

Reviewer 2 Report

Thank you for inviting me to review this manuscript, titled “Virtual Implicit Bias Reduction and Neutralization Training 2 (VIBRANT) for Mental Health Clinicians in Schools”.  This study propose to describe the iterative development and pilot feasibility trial of a brief, interactive, online implicit bias intervention known as Virtual Implicit Bias Re-duction and Neutralization Training (VIBRANT) with school-based mental health clini-cians.

The main question addressed by the research is relevant and interesting.

The specific issue or problem is defined.

The proposed objective is relevant, as well as the evidence provided and the accompanying discourse.

The analyzes used are justified and presented in an understandable way.

 Instruments with adequate reliability and validity psychometric properties were administered for this.

However, the manuscript would benefit from some small suggestions or changes. See specific suggestions below.

If possible I would try to shorten the title extension

It would be necessary to alphabetize the keywords in the abstract

Indicate something in the abstract regarding the type of design, study

Some statistics such as N should be written in italics both in the abstract

Expand the theoretical framework of the study with some more recent references in which the methodology used is used

If it is possible to indicate a theoretical model that more precisely integrates the variables of the study

Add more clearly the objective of this study at the end of the introduction

Expand the inclusion and exclusion criteria of the simple

State something about how the sample effect size has been controlled

Indicate something in procedure regarding the approval by the study of the Ethics Committee of the institution in which it has been carried out

Indicate something regarding the psychometric properties or use in other studies of the instrument  2.1.3.2. Implicit Bias Knowledge Quiz 187

Add an example of an item that is part of each of the instruments

Include a section with the description of the type of design used in the study

Some statistics such as p, t, M, SD, should be written in italics both in the abstract and within the text, in tables, etc.

Expand the section on discussion, future lines of research, study limitations and conclusions

Adjust all references to journal standards, eg title in italics, year in bold, etc. For example, the title and volume of the journal, etc. are not italicized. It is recommended to check an article from the latest issue of the journal

Round 2

Reviewer 1 Report

Thank you very much for your great effort to improve the manuscript. I consider that the results have been good. As far as I am concerned, there is just a question to keep on studying and making a review: the sample size. It would be interesting to include some explicit data to justify the representativity of the sample. 

Author Response

We greatly appreciate Reviewer 1’s continued thorough review and kind recognition of the work we put into improving the manuscript. We agree with this final suggestion on better justifying the representativeness of our study sample. Accordingly, we added 2 more citations of recent studies with school mental health clinicians in the United States demonstrating similar sample characteristics in terms of clinician demographic variables (e.g., race, gender, years of experience) on page 8, Line 344.

Reviewer 2 Report

The article has improved a lot with the suggestions made.

Author Response

We appreciate Reviewer 2’s rapid review and have double checked the manuscript for minor typographical errors.